Epigenetic identification of mitogen-activated protein kinase 10 as a functional tumor suppressor and clinical significance for hepatocellular carcinoma

Tang Liping 1
Zhu Shasha 2
Peng Weiyan 3
Yin Xuedong 3
Tan Cui 3
Yang Yaying 4 100500@cqmu.edu.cn
1 Department of Gastroenterology, the First Affiliated Hospital of Chongqing Medical University, Chongqing Medical University , Chongqing , China
2 The Center for Clinical Molecular Medical Detection, the First Affiliated Hospital of Chongqing Medical University , Chongqing , China
3 Chongqing Key Laboratory of Molecular Oncology and Epigenetics, the First Affiliated Hospital of Chongqing Medical University, Chongqing Medical University , Chongqing , China
4 Department of Pathology, Molecular Medicine and Cancer Research Center, Chongqing Medical University , Chongqing , China
Uversky Vladimir
Electronic publication date: 2021 Feb 2
Publication date: 2021
Volume: 9
Electronic Location ID: e10810
Received 2020 Apr 16; Accepted 2020 Dec 30
Copyright: © 2021 Tang et al.
Copyright year: 2021
Copyright holder: Tang et al.
License: This is an open access article distributed under the terms of the Creative Commons Attribution License, which permits unrestricted use, distribution, reproduction and adaptation in any medium and for any purpose provided that it is properly attributed. For attribution, the original author(s), title, publication source (PeerJ) and either DOI or URL of the article must be cited.
License URL: https://creativecommons.org/licenses/by/4.0/

Keywords: hepatocellular carcinoma, mapk10, Tumor suppressor gene, Methylation

Funding: National Natural Science Foundation Committee of China 81000907 This work was supported by the National Natural Science Foundation Committee of China under contract No. 81000907. The funders had no role in study design, data collection and analysis, decision to publish, or preparation of the manuscript.

==============================
Background

Mitogen-activated protein kinase 10 (Mapk10) is a member of the c-jun N-terminal kinases (jnk) subgroup in the MAPK superfamily, and was proposed as a tumor suppressor inactivated epigenetically. Its role in hepatocellular carcinoma (HCC) has not yet been illustrated. We aimed to investigate the expression and epigenetic regulation of mapk10 as well as its clinical significance in HCC.

Results

Mapk10 was expressed in almost all the normal tissues including liver, while we found that the protein expression of MAPK10 was significantly downregulated in clinical samples of HCC patients compared with these levels in adjacent normal tissues (29/46, P < 0.0001). Clinical significance of MAPK10 expression was then assessed in a cohort of 59 HCC cases, which indicated its negative expression was significantly correlated with advanced tumor stage (P = 0.001), more microsatellite nodules (P = 0.025), higher serum AFP (P = 0.001) and shorter overall survival time of HCC patients. Methylation was further detected in 58% of the HCC cell lines we tested and in 66% of primary HCC tissues by methylation-specific PCR (MSP), which was proved to be correlated with the silenced or downregulated expression of mapk10. To get the mechanisms more clear, the transcriptional silencing of mapk10 was reversed by pharmacological demethylation, and ectopic expression of mapk10 in silenced HCC cell lines significantly inhibited the colony formation ability, induced apoptosis, or enhanced the chemosensitivity of HCC cells to 5-fluorouracil.

Conclusion

Mapk10 appears to be a functional tumor suppressor gene frequently methylated in HCC, which could be a valuable biomarker or a new diagnosis and therapy target in a clinical setting.

Background

Hepatocellular carcinoma (HCC) is one of the most lethal malignancies in humans. It is the sixth most common cancer worldwide and the leading cause of deaths among patients with cirrhosis in Europe and the USA accompanied about 782,000 deaths per year (Lombardi et al., 2018; Bray et al., 2018). Since the patients with HCC are always diagnosed at the advanced stage and the efficacy of curative surgery or adjuvant therapy including systemic chemotherapy on advanced HCC remains poor (Zucman-Rossi et al., 2015; Chen et al., 2015; Kudo et al., 2017), it is particularly important to find reliable biomarkers in HCC early diagnosis and potential therapeutic targets.

Genetic and epigenetic alterations in regulatory genes are critical in the development of HCC, resulting in the imbalance between oncogenes and tumor suppressor genes (TSG) (Lee et al., 2016; Tischoff & Tannapfe, 2008). For example, promoter CpG methylation, one of the epigenetic alterations, which occurs frequently during tumor development and progression, causes the loss of TSG functions. Accumulating evidence demonstrated that the promoter of CpG methylation was essential in initiating carcinogenesis and the epigenomic changes, and thus, could be applied as a new biomarker for the diagnosis, in clinical trials of epigenetic drugs, and as an add-on to the drugs such as multikinases (Nakamura et al., 2019; Hardy & Mann, 2016; Ozen et al., 2013; Sceusi, Loose & Wray, 2011). Furthermore, it was identified that some valuable TSGs were frequently methylated in HCC and established a specific methylation profile that would provide novel reliable biomarkers for HCC management (Nishida et al., 2008).

MAPK10 (JNK3) encoded by the mapk10 gene is a member of JNKs, which were implicated in multiple signaling pathways including apoptosis, differentiation, and proliferation (Zhang & Dong, 2007; Davis, 2000). Previous studies showed that JNK contributed to the proliferation or survival of tumors; nevertheless, it might also act as a suppressor in some tumor cell types, indicating various roles of JNK in tumor progression (Kennedy & Davis, 2003). Hitherto, the genetic and mechanistic basis for the complex role of JNK in tumors is yet to be elucidated.

Loss of heterozygosity (LOH) at specific chromosomal regions in human tumors suggests the presence of candidate TSGs in the affected region. A deletion at 4q21.3 locus was identified in two esophageal carcinoma cell lines by high-resolution (1 Mb) array-comparative genomic hybridization (CGH); as a result, mapk10 was found to be localized within this deleted region (Ying et al., 2006). Other reports demonstrated that the frequent loss of expression of mapk10 in 10/19 human cell lines originated in brain tumors and the JNK3 signaling pathway mediated cell apoptosis in the central nervous system in jnk3-deficient mice (Yoshida et al., 2001; Yang et al., 1997). Moreover, the effect of Jnk deficiency on Ras-stimulated transformation resulted in a substantial increase in the number and growth of tumor nodules in vivo (Kennedy et al., 2003), which was consistent with the conclusion that mapk10 plays a role in tumor suppression. Furthermore, previous studies showed that the expression of mapk10 was frequently downregulated or silenced in several tumor cell lines such as lymphoma, breast cancer, gastric cancer, in which, the methylation of mapk10 promoter is correlated with its expression (Ying et al., 2006). Also, the ectopic expression of mapk10 markedly suppresses proliferation of breast cancer cells (Ying et al., 2006). In conclusion, these results suggested that mapk10 might function as a TSG to suppress the tumorigenesis with respect to epigenetics.

As far as we know, expression and role of mapk10 in HCC have not been illustrated. In our experiment, the expression of mapk10 was observed downregulated in HCC tissues, which dramatically correlated with the clinical characteristics and the overall survival (OS) of HCC patients. Furthermore, the experiments showed that promoter methylation was responsible for the mapk10 downregulation in HCC cell lines and tumor tissues, and the pharmacological demethylation and ectopic expression of mapk10 in silenced HCC cells could suppress the clonogenicity, induce the cell apoptosis, or enhance the sensitivity of HCC to 5-fluorouracil (5-FU). Overall our finding indicated that mapk10 be a functional tumor suppressor gene frequently methylated in HCC, suggesting that new therapy targeting mapk10 holds promise for HCC patients.

Materials and Methods

Tumor cell lines

A series of HCC cell lines (Hep3B, HepG2, huH1, huH4, huH6, huH7, PLC/PRF/5, SNU387, SNU398, SNU423, SNU449, and SNU475) were maintained in Cancer Epigenetics laboratory, Department of Clinical Oncology, PWH, The Chinese University of Hong Kong, Shatin, Hong Kong and cultured in RPMI or DMEM (Gibco, Gaithersburg, MD, USA) supplemented with 10% fetal bovine serum (FBS) (Gibco, Gaithersburg, MD, USA) and incubated at 37 °C in a humidified chamber containing 5% CO2.

Tissues of HCC

A total of 18 fresh primary HCC tissues were collected from patients who underwent hepatectomy for HCC in 2009 at the First Affiliated Hospital of Chongqing Medical University (Chongqing, China). Among them,10 has the matched non-tumor liver samples, obtained from at least 5 cm distant from the tumor. All of the fresh tissues were processed routinely for histopathological assessment and immediately immersed in liquid nitrogen after surgical resection.

Along with the paraffin-embedded tissues, a retrospective cohort of 59 patients with HCC was enrolled in the current study. These patients underwent hepatectomy at the First affiliated hospital of Chongqing Medical University (Chongqing, China) between June 2005 and December 2009. All patients were diagnosed with primary HCC by two pathological doctors, and there was no previous radio- or chemotherapy before surgery. The study was carried out in accordance with the Declaration of Helsinki and the protocol was approved by Ethics Committee of Chongqing Medical University (Chongqing, China) and written informed consent was obtained from each patient involved in the study.

Immunohistochemistry

IHC staining was carried out by using Ultrasensitive TMS-P (MXB, Fuzhou, China). The primary antibody against MAPK10 was obtained from Abcam (Abcam, Cambridge, MA, USA) and incubated the sections at a dilution 1:100. The staining intensity was scored as 0 (absent expression), 1 (weak expression), 2 (moderate expression), and 3 (strong expression) and the percentage of positive staining as 0 (0–9%), 1 (10–25%), 2 (26–50%) or 3 (51–100%) (Jiang et al., 2015). The final score was obtained by multiplying as the following: the staining intensity score × the percentage of positive staining and ranked from 0 to 9. So, the Mapk10 expression of HCC was defined as “−”, “+”, “++” and “+++” corresponding to score 0, 1–3, 4–6 and 7–9 respectively. For further clinical analysis, we divided the HCC into two subgroups: the “Mapk10-negative” subgroup (“−”) or “Mapk10-positive” subgroup "+−+++". The IHC staining score was performed in a double-blind manner by two pathologists to minimize the observational bias.

Semiquantitative RT-PCR

RT-PCR was conducted to detect the expression of Mapk10 in HCC cell lines. RT-PCR was performed for 32 cycles with hot-start Go-Taq (Promega, Madison, WI, USA) (Tao et al., 2002). The primers are listed in Table 1.

Table 1 PCR primer sequences and reaction conditions.

PCR	Primers	Size (bp)	TA (°C)	PCR cycles	
RT-PCR					
MAPK10	F3: 5′-cagctctctaaattgactcag-3′				
RR3: 5′-ccaatgttggttcactgcag-3′	248	55	32	
GAPDH	333: 5′-gatgaccttgcccacagcct-3′	
355: 5′-atctctgcccctctgctga-3′	304	55	23	
Methylation-specific PCR (MSP)					
Methylated	m3: 5′-cgagtagttttagcggttac-3′	160	60	40	
m5: 5′-aaaaccttctaacgcgaacga-3′	
Unmethylated	u3: 5′-tgtgagtagttttagtggttat-3′	163	58	40	
u5: 5′-caaaaccttctaacacaaacaa-3′	
Note:

TA, annealing temperature.

Western blot

Primary antibodies against Mapk10 (1:000, Abcam, USA) and PARP (1:000, Abcam, Cambridge, MA, USA) were used. At 48 h post-transfection, the cells were collected and lysed in RIPA lysis buffer (Beyotime, Haimen, China). Total protein was dissociated by SDS-PAGE, transferred onto PVDF membrane, and immunoblotted. The immunoreactive bands were exposed by the ECL detection system.

Methylation-specific PCR

After DNA bisulfite treatment, MSP was performed in the HCC cell lines as well as in the primary HCC tumors and adjacent non-tumor tissues as described previously (Tao et al., 2002). The MSP primers detecting the methylated or unmethylated alleles are MAPK10m3/MAPK10m5 and MAPK10u3/MAPK10u5, respectively (Table 1) (Ying et al., 2006). These primer pairs have been tested previously that they did not amplify any unbisulfited DNA. MSP was performed for 40 cycles with AmpliTaq-Gold (Applied Biosystems, Waltham, MA, USA) (Tao et al., 2002). YccB1, a cell line of breast carcinoma with hypermethylated MAPK10 which has been demonstrated in the previous literature was used as the positive control of MAPK10 methylation (Ying et al., 2006).

5-aza-2′-deoxycytidine (Aza) and Trichostatin A (TSA) treatment

A density of 1 × 105 HCC cells/mL were allowed to grow overnight. Then, fresh medium containing Aza at a final concentration of 10 mM (Sigma-Aldrich, St. Louis, MO, USA) was used to replace the culture medium. The cells were treated with Aza for 72 h, followed by the histone deacetylase inhibitor TSA at 100 nM for another 24 h. Finally, the cells were collected for methylation-specific PCR (MSP) and the gray value of bands was analyzed by Image J software.

Gene cloning and plasmid construction

The Mapk10 recombinant plasmids were purchased from GeneCopoeia (Rockville, MD, USA). These were termed as OmicsLink™ expression clone (EX-A1150-M29) and pcDNA3.0(+)-Flag-MAPK with the accurately confirmed sequence and orientation.

Subcellular localization

A density of 5 × 104 of Hep3B or HepG2 cells were planted on coverslips in a 6-well plate. Then, the cells were transfected with pcDNA3.0(+)-Flag-Mapk10 using Lipofectamine 2000 (Invitrogen, Carlsbad CA, USA). After 24 h, the cells were fixed with 4% paraformaldehyde phosphate buffer solution and incubated with anti-Flag M2 monoclonal antibody (Sigma, St. Louis, MO, USA), followed by FITC-conjugated rabbit anti-mouse IgG F(ab)2 antibody (Sigma, St. Louis, MO, USA). Hereafter, cell nuclei were stained with DAPI (Thermo, Waltham, MA, USA) and images were captured by Leica TCS SP2 AOBS confocal laser-scanning microscope (Mannheim, Germany).

Colony formation assay

1 × 105 Hep3B and HepG2 cells/well were seeded in a 12-well plate and transfected with Mapk10 plasmid or control vector, using Lipofectamine 2000. After 48 h, the cells were seeded into a 6-well plate, and subjected to G418 (0.4 mg/mL) for 10–14 days, with the medium refreshed every 2 days. The surviving colonies were enumerated after Giemsa staining. All the experiments were conducted three times in triplicate wells.

Apoptosis assay

The cell apoptosis and viability were evaluated using the Annexin V-PE Apoptosis Detection Kit I (BD Biosciences, San Jose, CA, USA) by flow cytometry. Apoptosis was always graded as viable, early apoptotic, necrotic and late apoptotic, which corresponds to Annexin V-/7-AAD-, Annexin V+/ 7-AAD-, Annexin V-/7-AAD+ and Annexin V+/7-AAD+. Hep3B and HepG2 cells were transfected with Mapk10 plasmid or the control vector and harvested after 48 h. The GFP-positive cells were selected, and the apoptotic status was assessed by Annexin V-PE and 7-AAD staining. Both early and late apoptotic cells were considered for relative apoptotic alterations. Apoptosis was also depended on the terminal deoxynucleotidyl transferase (TDT)-mediated dUTP-digoxigenin nick end labeling (TUNEL) assay. At 48 h post-transfection, the apoptotic cells were detected using the In Situ Cell Death Detection Kit POD (Roche, Burlington, NC, USA), wherein the DNA fragmentation was determined and visualized by a fluorescence microscope. All the experiments were repeated 3 times.

XTT proliferation assay

Hepatocellular carcinoma is resistant to systemic chemotherapy and 5-FU is one of the most common drugs for HCC. Herein, we assessed whether Mapk10 could enhance the chemotherapeutic effects of 5-FU on HCC cells. XTT Cell Proliferation Assay kit (Beyotime, Haimen, China) was used to determine the chemotherapy-induced cytotoxicity on HCC cells. Mapk10 or control vector was transfected into HCC cells, respectively and treated with 5-FU at different concentrations (0, 6.25, 12.5, 25, 50, 100, and 200 µg/mL) for 72 h. Subsequently, the relative number of viable cells between Mapk10-transfected group and control vector-transfected group was compared and expressed as a percentage using the following formula: cell viability (%) = A450 of cells treated with 5-FU/A450 of cells without 5-FU treatment. Three independent experiments were performed, each in duplicate.

Statistical analysis

The clinicopathological features in MAPK10-positive patients and MAPK10-negative patients were compared using the Pearson’s chi-squared test. Kaplan–Meier plots and log-rank tests were used for the survival analysis. Student’s t-test was used to compare the differences in the effect of MAPK10 expression on colony formation, cell apoptosis, and cell proliferation. SPSS 19.0 was used for data analysis. A P-value <0.05 was deemed as statistically significant.

Results

The protein expression of MAPK10 in clinical samples of HCC and its clinical significance in HCC patients

The cohort of 59 patients (Table 2) consisted of 77.9% males with a median age of 49.94 (range, 21–77) years. Chronic HBV carriers accounted for the majority (66.2%). The median AFP (α-fetoprotein) value was 5,181.8169 (range 2.8–96,401) ng/ml. Among all the samples, 46 pairs were eligible for the comparison of the expression of mapk10 between HCC and adjacent non-tumor tissues by IHC. All of the 59 patients with follow-up data were well qualified for the clinical correlation analysis and OS analysis, with the median follow-up for 13.0 (range 2–46) months.

Table 2 Clinicopathological correlation of Mapk10 protein expression in HCC.

Clinicopathological parameters	Number	Mapk10 expression	P value	
Features	(n = 59)	Negative	Positive		
Age					
≤60	49	18	31	0.442	
>60	10	5	5	
Sex					
Male	47	17	30	0.389	
Female	12	6	6	
HBsAg					
Negative	17	6	11	0.717	
Positive	42	17	25	
Serum AFP (ng/ml)					
≤400	30	8	22	0.05*	
>400	29	15	14	
Tumor size (cm)^:					
≤5	20	5	15	0.119	
>5	39	18	21	
Microsatellite nodules					
Absent	51	17	34	0.025*	
Present	8	6	2	
Cirhosis					
Absent	29	10	19	0.494	
Present	30	13	17	
PVTT					
Absent	47	18	29	0.834	
Present	12	5	7	
Tumor stage (AJCC)					
Stage I	23	3	20	0.001**	
Stage II	7	3	4	
Stage III	28	17	12	
Notes:

^ Tumor size was measured by the length of the largest tumor nodule.

* Statistically significant (P < 0.05).

** Statistically significant (P < 0.01).

PVTT: Portal vein tumor thrombus.

AJCC: American Joint Committee on Cancer.

As for 46 paired paraffin-embedded tissues, all of the adjacent non-tumor tissues show the positive expression of MAPK10, whereas 29 HCC tissues show the negative expression, the negative expression rate of MAPK10 between the HCC and the adjacent non-tumor tissues was significant (P < 0.0001) (Fig. 1). The clinicopathological correlation analysis showed that the negative expression of MAPK10 in HCC was significantly associated with higher serum AFP (p = 0.05), more microsatellite nodules (P = 0.025) and advanced tumor stage (P = 0.001) (Table 2).

Figure 1 Mapk10 protein expression in HCC specimens shown by immunohistochemistry.

Positive Mapk10 staining was observed in adjacent non-tumor tissue. Negative Mapk10 staining in HCC tissue. (A) Non-tomor tissue stained by H&E; (B) non-tomor tissue stained by IHC; (C) HCC tissue stained by H&E; (D) HCC tissue stained by IHC.

Downregulated MAPK10 expression may indicate poor prognosis in HCC patients

Prognostic significance of MAPK10 in HCC was also studied in this cohort of 59 patients with follow-up data, which indicated that the expression of MAPK10 was extremely correlative with OS of patients (log rank = 7.123, P = 0.008): the median OS time in MAPK10-negative subgroup (n = 23) and MAPK10-positive subgroup (n = 36) were 10.86 months (95% CI [5.927–15.792]) and 24.040 months (95% CI [17.511–30.568]) respectively (Fig. 2). These data suggested that MAPK10 could be an indicator for the prognosis of HCC patients.

Figure 2 Kaplan–Meier survival curves of HCC (n = 59) after gastrectomy.

The survival rate of patients in the group of negative MAPK10 expression was significantly lower than that of patients in the group of positive expression (log-rank test P = 0.008).

Expression profiling of Mapk10 in HCC cell lines

Previous studies found a deletion at 4q21.3 in two esophageal carcinoma cell lines and found that Mapk10 was located in this deleted region with a 633bp bidirectional promoter, which was a typical CpG island, as well as that Mapk10 was expressed in all normal adult tissues (Ying et al., 2006). However, semi-quantitative RT-PCR in our experiments revealed a frequently reduced or silenced expression of Mapk10 in HCC cell lines (67%, 8/12) (Fig. 3). Together, the above results showed that Mapk10 is a widely expressed gene but frequently disrupted in multiple HCC cell lines.

Figure 3 Mapk10 mRNA expression in HCC cell lines.

Frequent inactivation of Mapk10 by promoter CpG methylation

In fresh primary tumors, the Mapk10 promoter was frequently methylated in 12/18 (66%) HCC tissues, while reduced or no methylation was detected in the adjacent non-tumor tissues (Fig. 4A), suggesting that the methylation of Mapk10 is tumor-specific. Furthermore, we detected Mapk10 methylation in HCC cell lines using MSP and found that the methylation of Mapk10 is associated with the dysregulated or silenced expression of the gene shown in RT-PCR (Figs. 4B and 4C).

Figure 4 Promoter CpG methylation detection of Mapk10.

(A) Mapk10 methylation in HCC and adjacent non-tumor tissues. (B) Mapk10 methylation in HCC cell lines. (C) Mapk10 gene expression in HCC cell lines (MSP: methylation-specific PCR; M: methylated alleles; U: unmethylated alleles).

Pharmacological and genetic demethylation restores Mapk10 expression

Two HCC cell lines, Hep3B and HepG2, were subjected with DNA methyltransferase inhibitor Aza combined with histone deacetylase inhibitor Trichostatin A (TSA). The treatment restored the Mapk10 expression, accompanied by decreased methylated promoter alleles (HepG2) or increased unmethylated alleles (Hep3B) (Fig. 5). The above results indicated that promoter methylation directly mediated the transcriptional silencing of Mapk10.

Figure 5 Mapk10 expression of HCC cell lines after treatment with TSA and 5-Aza (MSP: Methylation-specific PCR; M: methylated alleles; U: unmethylated alleles).

The frequent silencing of Mapk10 by hypermethylation in HCC cell lines rendered Mapk10 as a putative tumor suppressor. Furthermore, we evaluated the growth characteristics of the cells overexpressing Mapk10 by colony formation assay. HepG2 and Hep3B cell lines with silenced Mapk10 were transfected with Mapk10 recombinant plasmids, and the number of colonies was enumerated after 10–14-day culture under G418 selection. The subcellular localization of MAPK10 was examined by indirect immunofluorescence staining, which indicated that Flag-tagged MAPK10 was localized in the nucleus after the transfection into HCC cells (Fig. 6A) and the western blot detection of MAPK10 protein expression also confirmed the success of transfection (Figs. 6B and 6C). The ectopic expression of Mapk10 markedly decreased the colony formation efficiency of Hep3B and HepG2 cells (34% and 40% respectively, **P < 0.01), indicating the growth inhibitory activity of Mapk10 (Figs. 6D and 6E).

Figure 6 MAPK10 inhibits the colony formation of HCC cells.

(A) The subcellular localization of MAPK10 was observed after HCC cell lines were transfected with Mapk10 recombinant plasmids. (B) MAPK10 protein expression detected by WB. (C) Analysis of MAPK10 protein expression. (D) Colony formation detection. (E) Colony formation efficiency (*P < 0.05, **P < 0.01).

Ectopic expression of Mapk10 induces apoptosis

To explore the mechanism of the tumor suppression by Mapk10, we performed apoptosis assay using Annexin V-PE and 7-AAD double staining. The percentage of Annexin V (+/−) and 7-AAD (+/−) cells in GFP-positive cells by flow cytometry was determined. In Hep3B cells and HepG2 cells, the ectopic Mapk10 expression resulted in a significant increase in the number of apoptotic cells as compared to the control (P < 0.05) (Fig. 7A). Furthermore, the apoptotic induction was assessed by TUNEL assay at the individual cell level. Green fluorescent signals, characteristic of cell apoptosis, were presented in Mapk10-transfected cells but less in control vector- transfected cells (Fig. 7B). Increased protein level of cleaved-PARP was also shown in HepG2 and Hep3B cells after MAPK10 transduction (Fig. 7C).

Figure 7 Ectopic expression of MAPK10 induces apoptosis of HCC cell lines.

(A) Apoptosis assay using Annexin V-PE and 7-AAD double staining, *P < 0.05. (B) TUNEL assay. (C) Protein level of cleaved-PARP detected by WB after the transduction of Mapk10.

Mapk10 enhances the chemosensitivity of HCC cells to 5-FU

The efficacy of chemotherapeutic drugs is dependent on their ability to trigger apoptosis (Jiang et al., 2015). Since our studies showed that Mapk10 induces apoptosis in HCC cells, we speculated the enhanced chemotherapy effects of 5-FU on HCC cells. Notably, the cell viability in Mapk10-transfected group was significantly inhibited as compared to that in the control group (Fig. 8A). Furthermore, the IC50 of HepG2 cells for 5-FU was significantly reduced after Mapk10 transfection, which was 16.253 µg/mL in the Mapk10-transfected group as compared to 251.356 µg/mL for the control group (Fig. 8B).

Figure 8 MAPK10 enhances the chemosensitivity of HCC cells to 5-FU.

(A) Cell viability assay of Hep3B after transfected with Mapk10. (B) Cell viability assay of HepG2 after transfected with Mapk10 (**P < 0.01).

Discussion

In the present study, we found that the expression of Mapk10 is frequently silenced or downregulated in most of the paraffin-embedded HCC tissues (63%) when compared with the adjacent non-tumor tissues. The clinical analysis of the paraffin-embedded tissues also revealed that the negative expression of Mapk10 was significantly associated with higher serum AFP, more tumor microsatellite nodules, advanced tumor stage, and the shorter OS of HCC patients. Moreover, hypermethylation of Mapk10 promoter was further detected in most of these primary HCC tumors (66%), and it was also demonstrated that the hypermethylation is correlated with the silenced gene expression of Mapk10 in several HCC cell lines. Conversely, Mapk10 hypermethylation was reduced in paired non-tumor tissues, suggesting a role of Mapk10 as a candidate TSG in the pathogenesis of HCC. The expression of reactivated Mapk10 could be observed after treatment with the demethylating reagent in silenced HCC cells, indicating that DNA hypermethylation directly mediated the inactivation of Mapk10. Furthermore, the ectopic expression of Mapk10 in silenced HCC cells dramatically inhibited the tumor cell colony formation, induced the cell apoptosis, and enhanced the chemosensitivity of HCC cells to 5-FU. Thus, Mapk10 was suggested to function as a TSG inactivated epigenetically for HCC and might act as a clinical biomarker for early diagnosis and therapy target.

The epigenetic inactivation of TSGs by the hypermethylation of promoter has been recognized as an alternative and critical mechanism in tumorigenesis. Rather than changing the genetic information, DNA methylation only changes the readability of the DNA and results in the inactivation of genes by subsequent suppression of mRNA transcription. As we know, TSG protects the cells from malignant transformation by mechanisms such as apoptosis or cell cycle regulation (Hayslip & Montero, 2006). Thus, the aberrant hypermethylation of the promoter and epigenetic silencing of TSG affect every step of tumor progression (Jones & Baylin, 2002; Luo, Hajkova & Ecker, 2018; Dor & Cedar, 2018). Currently, the transcriptional silencing of TSGs associated with DNA methylation has been shown to be a common epigenetic event in several tumors, including hematologic malignancies, nasopharyngeal carcinomas, gastroenterological neoplasia, esophageal carcinomas, and breast cancers (Jones & Baylin, 2002; Tao & Chan, 2007; Rashid & Issa, 2004; Xiang et al., 2013; Barbano et al., 2013). The identification of potential TSGs and specific methylation profiles of TSGs for every specific tumor might be valuable for the diagnosis of tumor and prediction of the prognosis (Kazanets et al., 2016).

So far, the studies in HCC found some epigenetic silenced TSGs, such as p16, E-cadherin, hMLH1, SOCS1, SOCS, and RASSF1A, which play a role in proliferation, apoptosis, cell adhesion, invasion, and DNA repair (Tischoff & Tannapfe, 2008). Compared to several other tumors, the prognosis of HCC is poor, and the underlying molecular pathogenesis, including the epigenetic mechanism, is less understood. Most of the HCCs arise on the background of chronic liver disease and chronic hepatitis B virus (HBV), hepatitis C virus (HCV), aflatoxin, alcohol consumption, and hemochromatosis constitute the major factors associated within HCC initiation (Herath, Leggett & MacDonald, 2006). Reportedly, the aberrant methylation of TSG in HCC occurs in a genetic- and disease-specific manner and could be observed not only in advanced tumors but also in premalignant conditions of HCC (Tischoff & Tannapfe, 2008; Nishida et al., 2008). Also, studies revealed that HCC and precancerous lesions might present epigenetic signals correlated with the specific risk factors and tumor progression stage, indicating that the epigenetic changes in HCC might be potential targets for biomarker discovery or future therapeutic strategies (Herceg & Paliwal, 2011).

Mapk10 (JNK3) is a third member of JNKs, which belongs to MAP kinases, implicated in critical physiological processes, including proliferation, apoptosis, and differentiation (Davis, 2000; Kennedy & Davis, 2003). Numerous studies have confirmed that JNK acts as a suppressor of Ras transformation and contributes towards apoptosis in tumors. For example, the activation of the mitochondrial apoptotic pathway in many cell types and the interferon-alpha (IFN-α)-induced apoptosis in B-cell lymphoma were reported (Yanase et al., 2005). Nevertheless, it might also enhance the proliferation and survival responses of some tumors (Kennedy & Davis, 2003), thereby rendering it as a putative TSG. Thus, the strategies of tumor prevention and therapy necessitate further in-depth investigation. Unlike JNK1 and JNK2, Mapk10 (JNK3) is not expressed ubiquitously and performs non-redundant functions (Yang et al., 1997). Frequent loss of Mapk10 expression has been detected in a panel of tumor cell lines (Ying et al., 2006; Yoshida et al., 2001), together with the proapoptotic functions observed (Yang et al., 1997; Kennedy et al., 2003), suggesting its potential role as a TSG. Previously, it was identified that Mapk10 (JNK3) could be inactivated in multiple lymphomas and carcinomas by epigenetic methylation, albeit only few studies have addressed this concern (Ying et al., 2006). In the present study, we focused on HCC. Mapk10 (JNK3) was shown to be silenced or downregulated in a majority of the HCC cell lines or primary HCC tumor samples examined, while the demethylating reagent could restore its expression in silenced HCC cells. Further steps indicated that the ectopic expression of Mapk10 in silenced HCC cells significantly inhibited the tumor cell growth or induced the apoptosis. So our results demonstrated that Mapk10 served as a TSG in HCC and could be inactivated by promoter methylation in tumorigenesis, which might be one of the mechanisms in HCC progression. However, due to the complexity of gene expression regulation, the additional mechanisms beside methylation may be involved in the regulation of Mapk10 expression and needed for further explored.

As we know, chemoresistance was facing challenge with respect to the treatment of HCC patients (Lohitesh, Chowdhury & Mukherjee, 2018). The efficacy of chemotherapeutic drugs has been speculated to depend on the ability to trigger apoptosis (Chen, Chan & Guan, 2011). In our research, Since MAPK10 could induce apoptosis in these HCC cells; we speculated that it might also enhance the efficacy of the chemotherapeutic drugs on HCC. Herein, the chemotoxic effects of 5-FU on HCC cells were compared between the Mapk10-transfected group and the control vector-transfected group. Consequently, significant enhancement of chemosensitivity of HCC cells to 5-FU was observed in Mapk10-transfected group, indicating ectopic expression of Mapk10 as a novel therapeutic assistant for drug resistance in HCC. Finally, the clinical correlation analysis found that the silenced expression of Mapk10 in HCC was dramatically correlative with the advanced tumor stage and poor prognosis of HCC patients. This phenomenon supported that Mapk10 expression might be a valuable molecular biomarker for HCC in a clinical setting.

Conclusion

In summary, Mapk10 is functions as a tumor suppressor and frequently inactivated by promoter methylation in HCC. The downregulated expression of Mapk10 caused by hypermethylation in HCC tumors is correlated with the advanced tumor stages and poor prognosis of patients, which hinted that the epigenetic inactivation of Mapk10 is one of the major mechanisms in tumorigenesis of HCC, along with the evidence that Mapk10 is a general TSG for HCC. The high incidence of epigenetic inactivation of Mapk10 in HCC rendered hypermethylation as a reliable molecular biomarker and a potential epigenetic therapeutic target in HCC management.

Supplemental Information

Supplemental Information 1 Original blots.

Click here for additional data file.

Supplemental Information 2 PCR primer sequences and reaction conditions.

Click here for additional data file.

We greatly thank Professor Qian Tao (Cancer Center, Department of Clinical Oncology, PWH, The Chinese University of Hong Kong, Shatin, Hong Kong) for his kindly help and guidance in the experimental design and methodology.

Additional Information and Declarations

Competing Interests

Author Contributions

Human Ethics

Data Availability

The authors declare that they have no competing interests.

Liping Tang conceived and designed the experiments, performed the experiments, analyzed the data, prepared figures and/or tables, authored or reviewed drafts of the paper, and approved the final draft.

Shasha Zhu performed the experiments, authored or reviewed drafts of the paper, and approved the final draft.

Weiyan Peng conceived and designed the experiments, prepared figures and/or tables, and approved the final draft.

Xuedong Yin performed the experiments, authored or reviewed drafts of the paper, and approved the final draft.

Cui Tan analyzed the data, prepared figures and/or tables, and approved the final draft.

Yaying Yang performed the experiments, prepared figures and/or tables, authored or reviewed drafts of the paper, and approved the final draft.

The following information was supplied relating to ethical approvals (i.e., approving body and any reference numbers):

The study was carried out in accordance with the Declaration of Helsinki and the protocol was approved by Ethics Committee of Chongqing Medical University (Chongqing, China) (2019-060).

The following information was supplied regarding data availability:

Data and uncropped blots are available in the Supplemental Files.

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
