# Peer review of "Epigenetic identification of mitogen-activated protein kinase 10 as a functional tumor suppressor and clinical significance for hepatocellular carcinoma"

_PeerJ, doi:10.7717/peerj.10810_

## Round 0.1 · original submission · Major Revisions

Please carefully review the comments from the peer reviewers, particularly those that relate to experimental controls, technical rigor and accuracy of reporting.

Reviewer 1 ·

Basic reporting

1. English language should be improved and typos are present in the main text; English proofreading is recommended.

2. Thanks to the authors for uploading the raw data; however, some raw data included as supplementary material are not commented in the main text; see the section "General comments for the author"

3. Overall, the results as presented in this manuscript do not fully support the initial hypotheses

Experimental design

Some experiments were not performed respecting the technical standards for the used experimental procedures, please see the comments in the "General comments for the author"

Validity of the findings

Overall, the data provided in this manuscript are not very robust to fully support the conclusions as stated (please see in "General comments for the author")

Additional comments

In the current study, Tang and colleagues have focused their research on investigating the expression and the role of MAPK10 in human HCC. By performing immunohistochemistry analysis, the authors could detect a downregulation of MAPK10 protein in HCC samples when compared to the adjacent non-tumor tissues; more importantly, negative expression of MAPK10 in HCC was associated to high AFP levels, advanced tumor stage and shorter overall survival of HCC patients following liver resection. Further experiments showed that MAPK10 expression is reduced in human HCC by promoter methylation. MAPK10 overexpression in vitro led to inhibition of colony formation and apoptosis induction. The authors concluded that MAPK10 might represent a new therapeutical target in a clinical setting.
Overall, the data as presented in this manuscript do not fully support the conclusions stated by authors; therefore, the manuscript needs to be revised in many aspects and complemented with new experiments in order to be considered for a new revision.
Here the major points, which need to be addressed:
1. The authors performed semi-quantitative RT-PCR for detecting MAPK10 expression in human HCC cell lines as shown in Fig.3; however, in the image presented in the figure it is possible to see additional upper bands, which might indicate not specific primer binding to any other sequence; could the authors comment on that? It would be helpful to include in the same analysis any type of positive control (human normal hepatocyte?). Additionally, detection of protein levels in the same cell lines is necessary to complement the expression data, considering also the data presented in Fig.4B (see comment below).
2. In Fig.4A/B positive controls for unmethylated and methylated DNA must be included in the assay in order to determine the quality of the bisulfite conversion and identify artifacts such as mispriming. In addition, the authors should add any comment about the detection of unmethylated MAPK10 in all the HCC samples used in order to better integrate these results with the IHC results; indeed, are additional mechanisms involved in MAPK10 expression regulation? The same comment is valid for the results presented in Fig.4B; do they authors know or speculate why MAPK10 promoter in the HuH4 is unmethylated but there is no expression of MAPK10?
3. The authors stated in the main text that cells were treated with 5-Aza and TSA (line 241-242), but in the legend for figure 5 (line 474) the only treatment specified is with TSA and in the image labels is again A+T; which is the treatment performed in those cells? In the case that Hep3B and HepG2 cells were treated with Aza and TSA, no changes were observed at MAPK10 methylation level when compared to the untreated control cells in the MSP results, especially for Hep3B, meaning that promoter methylation is not playing a role in the regulation of MAPK10 expression, how to explain these results?
4. In Fig.6A, Flag signal overlaps mostly with DAPI especially in HepG2 cells, which does not support the conclusions drawn by the authors. Any comment for these results? Analysis of MAPK10 protein in fractionated cytoplasmatic and nuclear protein lysates would be helpful for describing the cellular localization of MAPK10.
5. The technical approach used for performing the colony formation is not right; indeed, it would have been more appropriated to transfect first the cells with MAPK10 and control plasmid and select the cells, then seed the same number of cells after selection for each condition and let them to grow for 10-14 days, leaving out with this approach the unwanted difference in transfection efficiency of different plasmids (empty vector and vector with the gene insert), therefore difference in number of cells which can form colonies.
6. Apoptosis data presented in Fig. 7A should be confirmed by detection of cleaved PARP and cleaved Caspase 3 proteins.
7. English language should be improved and typos are present in the main text; English proofreading is recommended.
8. Excel file for raw data includes some data not displayed or commented in the main text (figure 7G?); can the authors explain these results?



Minor points:
1. The panel C in Fig.6 shows the relative MAPK10 protein expression; do the data include three biological replicates? The raw data or images for the three replicates should be supplied.

·

Basic reporting

1. Please see Line 53. The word “finding” should be “find”.
2. Please describe the known tumor suppressor genes frequently methylated in HCC. Compare those known tumor suppressor genes with MAPK10 or include them in this study as positive controls.
3. Are there any known tumor suppressor genes which belong to JNK family implicated in HCC?
4. In Line 75-77, the provided reference no.18 indicates JNK3 deficiency induced cell apoptosis in the brain, which is opposite to what the authors observed in the current study. Please explain why MAPK10 (JNK3) overexpression induced cell apoptosis in the current study.

Experimental design

1. Please provide the reference of the scoring method for the IHC staining of HCC liver sections.
2. Please explain why using histone deacetylase inhibitor TSA in this study. Results of DNA methyltransferase inhibitor Aza treated alone can better elucidate the effect of MAPK10 methylation in HCC cell lines.
3. Please provide the reference to support the methylation sites detected in this study (Supplementary Table 1).

Validity of the findings

1. It’s hard to see the association between MAPK10 expression and methylation in HCC cell lines in Figure 4B. Please add graphs to show the association clearly.
2. Please add colony forming assay, apoptosis assay, and cell viability assay for HCC cell lines after Aza treatment to better address the causal effect of MAPK10 methylation in HCC progression.
3. In Figure 3, please provide a non-cancer hepatic cell sample as a control for MAPK10 expression.
4. Please add an internal control to Figure 4A.
5. In figure 5, the methylated MAPK10 expression seems to be unchanged after A+T treatment in Hep2B cell line. Please explain.

Additional comments

I commend the authors for their extensive data set, compiled over many years of detailed fieldwork. In addition, the manuscript is clearly written in professional, unambiguous language. There are some issues which should be improved upon before Acceptance.

---

## Round 0.2 · Major Revisions

Thank you for the edits to your manuscript, as you can see the reviewers have some existing concerns remaining. Please note the apparent discrepancy between methylation and expression, and the concerns raised by reviewer 1 about the interpretation of the results from 5-Aza and TSA treatment.

Reviewer 1 ·

Basic reporting

1. English language has been improved

2. additional literature references should be provided (see comments)

3. The new results and the new comments are not exhaustive for supporting all their conclusions.

Experimental design

Please see the comments

Validity of the findings

The validity of the findings (in particular in Fig.5) is not fully supported by the data and the comments provided in the revised version of this manuscript by the authors.

Additional comments

1. Previous comment: 2. In Fig.4A/B positive controls for unmethylated and methylated DNA must be included in the assay in order to determine the quality of the bisulfite conversion and identify artifacts such as mispriming. In addition, the authors should add any comment about the detection of unmethylated MAPK10 in all the HCC samples used in order to better integrate these results with the IHC results; indeed, are additional mechanisms involved in MAPK10 expression regulation? The same comment is valid for the results presented in Fig.4B; do they authors know or speculate why MAPK10 promoter in the HuH4 is unmethylated but there is no expression of MAPK10?
"We included the YccB1, a TSG hypermethylated in the breast cancer, in our study as positive controls. At the same time, we also included the negative control in it. We are sorry that these controls were not shown in the previous versions and we have already corrected the Fig.4A/B in our revision. Among the 18 fresh primary HCC tissues, 15 of them underwent immunohistochemical staining at the same time and we found that 11 of them lost expression of MAPK10 and the corresponding Mapk10 promoter methylation was positive. We can see a band indicating the low expression of MAPK10 in the HuH4 RT-PCR results, but there is no obvious methylation detected in its promoter. We think some additional mechanisms maybe involved in Mapk10 regulation."

New comment: The authors included YccB1 as positive control gene for their analysis; could the authors please provide explanation about this gene and references? NCBI gene database doesn´t retrieve any result about this gene. In addition, the section of M&M should be updated accordingly. Additionally, the correct controls for the MSP are 100% methylated DNA and 100% unmethylated DNA samples (commercially available), which the authors have to include in their experiments.


2. Previous comment: 3. The authors stated in the main text that cells were treated with 5-Aza and TSA (line 241-242), but in the legend for figure 5 (line 474) the only treatment specified is with TSA and in the image labels is again A+T; which is the treatment performed in those cells? In the case that Hep3B and HepG2 cells were treated with Aza and TSA, no changes were observed at MAPK10 methylation level when compared to the untreated control cells in the MSP results, especially for Hep3B, meaning that promoter methylation is not playing a role in the regulation of MAPK10 expression, how to explain these results?
"We are sorry that there is a mistake in the legend (line 474), the cells were actually treated with 5-Aza and TSA in our experiment and we have already corrected that in our revision. In addition, the MSP result showed that there was the decrease in Mapk10 methylation level for HepG2 and the increase in Mapk10 unmethylation lever for Hep3B after treatment with 5-Aza and TSA, the inhibitors of methylation, which indicated that the transcriptional silence of Mapk10 for these two cell lines was caused by promoter methylation."

The answer provided by the authors doesn´t explain the results reported in Fig. 5; cell line treatment with 5-Aza and TSA should lead to decreased methylation and increased unmethylated levels in a similar way in both cell lines by MSP detection and the MSP results should be in line with the PCR results showed in the upper panel of Fig.5. Since the results presented in Fig.5 are not convincing, the authors should provide an additional proof that methylation and histone acetylation are involved in the regulation of MAPK10 expression in HCC (pyrosequencing?)


3. Which antibody was used for detection of PARP?

·

Basic reporting

Thanks to the authors for the correction of typos.

Experimental design

Thanks to the authors for adding the references.

Validity of the findings

1. Please explain why mRNA expression of MAPK10 is not negatively correlated with its methylation in each HCC cell line shown in Figure 4B.

---

## Round 0.3 · Minor Revisions

Please address remaining issues pointed by the reviewer and amend your manuscript accordingly.

Reviewer 1 ·

Basic reporting

Still some English typos or ambigous sentences are present in the "Discussion" section

Experimental design

Please see "General comments"

Validity of the findings

See "general comments" section

Additional comments

Thanks to the authors for replying to comment 1 and 3.
Regarding the comment 2, it is more matter of 5-Aza concentration and time treatment; the authors should add more details about the method used for interpretation of results (thickness of band, M or U) in the corrisponding section in materials and methods.
Additionally, the authors should add a sentence in the discussion that additional mechanisms beside methylation are involved in the regulation of Mapk10 expression.

---

## Round 0.4 · accepted · Accept

Thank you for addressing remaining issues. The amended manuscript is acceptable now. Congratulations!

Happy Holidays!